# Structural insights into the potency and selectivity of covalent pan-FGFR inhibitors

Lingzhi Qu[1,4], Xiaojuan Chen[1,4], Hudie Wei [1✉], Ming Guo[1], Shuyan Dai[1], Longying Jiang[1,2], Jun Li[1], Sitong Yue[1], Zhuchu Chen[1] & Yongheng Chen [1,3✉]

FIIN-2, TAS-120 (Futibatinib) and PRN1371 are highly potent pan-FGFR covalent inhibitors targeting the p-loop cysteine of FGFR proteins, of which TAS-120 and PRN1371 are currently in clinical trials. It is critical to analyze their target selectivity and their abilities to overcome gatekeeper mutations. In this study, we demonstrate that FIIN-2 and TAS-120 form covalent adducts with SRC, while PRN1371 does not. FIIN-2 and TAS-120 inhibit SRC and YES activities, while PRN1371 does not. Moreover, FIIN-2, TAS-120 and PRN1371 exhibit different potencies against different FGFR gatekeeper mutants. In addition, the co-crystal structures of SRC/FIIN-2, SRC/TAS-120 and FGFR4/PRN1371 complexes reveal structural basis for kinase targeting and gatekeeper mutations. Taken together, our study not only provides insight into the potency and selectivity of covalent pan-FGFR inhibitors, but also sheds light on the development of next-generation FGFR covalent inhibitors with high potency, high selectivity, and stronger ability to overcome gatekeeper mutations.

[1] Department of Oncology, NHC Key Laboratory of Cancer Proteomics, State Local Joint Engineering Laboratory for Anticancer Drugs, Xiangya Hospital, Central South University, Changsha 410008 Hunan, China. [2] Department of Pathology, Xiangya Hospital, Central South University, Changsha 410008 Hunan, China. [3] National Clinical Research Center for Geriatric Disorders, Xiangya Hospital, Central South University, Changsha 410008 Hunan, China. [4] These authors contributed equally: Lingzhi Qu, Xiaojuan Chen. ✉email: hudiewei18@163.com; yonghenc@163.com

The human fibroblast growth factor receptor (FGFR) family consists of four members (FGFR1-4)[1–3]. Through sensing FGF ligands and triggering downstream signaling pathways, FGFRs play critical roles in embryonic development, tissue homeostasis, metabolism and many other physiological processes[4,5]. Dysregulation of FGF/FGFR signaling pathways are thought to be oncogenic[6,7]. Aberrant expression of FGFRs, including gene mutation, amplification or overexpression, has been reported in a variety of human tumors, and associated with poor prognosis, metastasis and drug resistance[8,9]. FGFRs are therefore considered as attractive targets for cancer therapy[10].

Small-molecule tyrosine kinase inhibitors provide promising prospects for the treatment of relevant tumors associated with abnormal FGFR expression[11]. Multiple FGFR inhibitors have entered clinical trials, and Erdafitinib and Pemigatinib were approved by the FDA for the treatment of urinary bladder cancers[12] and cholangiocarcinoma[13], respectively. Owing to the conserved ATP pocket across human kinases, a great challenge in the development of FGFR inhibitors is associated with low selectivity and off-target-related clinical toxicities. Notably, covalent inhibitors have attracted increasing attention in recent years owing to their potential of improved selectivity, stronger potency and overcoming gatekeeper mutations-related resistance[14,15].

FIIN-2[16], PRN1371[17] and TAS-120 (Futibatinib)[18,19] are three irreversible pan-FGFR covalent inhibitors. PRN1371 and TAS-120 have entered into clinical investigation[20]. These three inhibitors possess an electrophilic acrylamide group designed to undergo Michael addition reaction and achieve covalent binding with a p-loop cysteine of FGFRs (Fig. 1a, b). Out of 518 human kinases, only 9 kinases share this p-loop cysteine (Fig. 1a), suggesting that these inhibitors might be highly selective. Among these 9 kinases, 4 of them (FGFR1-4) belong to FGFR family, and 3 of them (SRC, YES, FGR) belong to the SRC family kinases (SFKs). Until now, it remains unclear whether these three pan-FGFR inhibitors can potently inhibit SRC family kinases, leading to unwanted side effects such as skin eruption, fluid retention, and skin or hair depigmentation[21].

Herein, we report that FIIN-2 and TAS-120 form covalent adducts with SRC, and inhibit SRC and YES activities, whereas PRN1371 does not. We also find that these three inhibitors exhibit different potencies against different FGFR gatekeeper mutants. In addition, we also determined crystal structures of SRC/FIIN-2, SRC/TAS-120, and FGFR4/PRN1371 complexes with high resolution x-ray diffraction data. These three crystal structures detail the binding modes of these inhibitors, and provide structural basis for kinase targeting and overcoming gatekeeper mutations.

## Results

### Binding modes of FIIN-2, TAS-120 and PRN1371 to FGFR1 and SRC.
To test whether FIIN-2, TAS-120 and PRN1371 form covalent protein-inhibitor adducts with SRC protein, we carried out mass spectrometry, using FGFR1 as a positive control. All three inhibitors exhibited the expected shift in the mass of FGFR1 protein (Fig. 1c). When we incubated SRC with these three inhibitors, mass shifts were observed for FIIN-2 and TAS-120, however, no mass shift was observed for PRN1371 (Fig. 1d). When the p-loop cysteine of SRC and FGFR1 proteins was mutated to alanine, the molecular weight of the SRC and FGFR1 proteins did not change after incubation with these inhibitors (Supplementary Fig. 1). These results suggest that FIIN-2 and TAS-120 form covalent adducts with the p-loop cysteine of SRC, while PRN1371 does not.

### Inhibitory potency of FIIN-2, TAS-120 and PRN1371 against FGFRs and SFKs.
Previous studies have shown that all these three inhibitors are potent FGFR inhibitors[16,17,22], however, their potencies against FGFR proteins have not been compared side by side in the same experiment. We used kinase assay to compare their potency against all four FGFR proteins. All three inhibitors showed strong potencies against FGFR1-3 with $IC_{50}$ values below 10 nM (1.6–7.8 nM), and exhibited slightly weaker potencies against FGFR4 with $IC_{50}$ values of 12–31 nM (Fig. 2). These data are consistent with previous measurements[16,18,23]. These results indicate that these three inhibitors have very similar potencies against FGFR proteins.

Then we also used kinase assay to test whether these inhibitors are potent against SFK proteins. We chose three SFK proteins for the experiment. Among the three SFK proteins, SRC and YES proteins have the analogous p-loop cysteine, whereas HCK does not (Fig. 1a). FIIN-2 was moderately potent against SRC (330 nM) and YES (365 nM), and TAS-120 showed weak inhibition of SRC (1673 nM) and YES (1626 nM). Notably, PRN1371 only showed negligible potency against SRC and YES (Fig. 3a, b). When the p-loop cysteine of SRC was substituted with alanine (C277A), FIIN-2 and TAS-120 lost their potency against SRC (Fig. 3c). In addition, all three inhibitors showed negligible potency against HCK, which possesses a glutamine instead of a cysteine in the p-loop (Fig. 3d). These results suggest that PRN1371 is a more selective FGFR inhibitor than the other two inhibitors.

### Potency of FIIN-2, TAS-120 and PRN1371 against FGFR gatekeeper mutants.
Acquired resistance to kinase inhibitors is a major obstacle to long-term cancer treatment, especially when caused by gatekeeper mutations[11]. Known FGFR gatekeeper mutations include FGFR1(V561M), FGFR2(V564I/F), FGFR3(V555M), and FGFR4(V550L/M)[13]. We used kinase assay to compare the inhibitory potency of FIIN-2, TAS-120 and PRN1371 against gatekeeper mutations of FGFRs. As shown in Table 1, different inhibitors behaved differently against different FGFR gatekeeper mutants, and their potency against FGFR gatekeeper mutants was much lower than that against wild-type FGFRs. For FGFR1(V561M) mutant, all three inhibitors showed about 30-fold decreased potency with $IC_{50}$ values of 84–224 nM, with PRN1371 being the most effective (Fig. 4a). For the FGFR2 (V564F) mutant, TAS-120 and FIIN-2 showed a 20-fold and 95-fold decrease in potency with IC50 values of 52 nM and 276 nM, respectively (Fig. 4b). PRN1371 only showed weak inhibition on the mutant (Fig. 4b). For the FGFR3(V555M) mutant, the potency of FIIN-2, TAS-120 and PRN1371 was reduced by 15-, 50- and 320-fold, respectively, with IC50 values of 97–583 nM (Fig. 4c). For FGFR4(V550L), TAS-120 and FIIN-2 exhibited a 7- and 12-fold decrease in potency with $IC_{50}$ value of 90 and 255 nM, respectively, while PRN1371 only showed little effect on the mutant (Fig. 4d).

### Structural basis for SRC targeting by FIIN-2 and TAS-120.
In order to study the structural basis for SRC targeting by FIIN-2 and TAS-120, we determined the crystal structures of FIIN-2 and TAS-120 in complex with SRC at resolutions of 2.1 Å and 2.7 Å, respectively. Data collection and structure refinement statistics are presented in Supplementary Table 1. In the SRC/FIIN-2 structure, a covalent bond between the acrylamide group of FIIN-2 and the sulfhydryl group of Cys277 is observed. In addition to the covalent bond, FIIN-2 forms two hydrogen bonds with the backbone atoms of Met341, and forms a hydrogen bond with Cys277 (Fig. 5a). The interactions between FIIN-2 and SRC are further stabilized by extensive van der Waals contacts (Supplementary Fig. 2a). These interactions are similar to those of the FGFR4/FIIN-2 complex (PDB ID: 4QQC) (Fig. 5b,

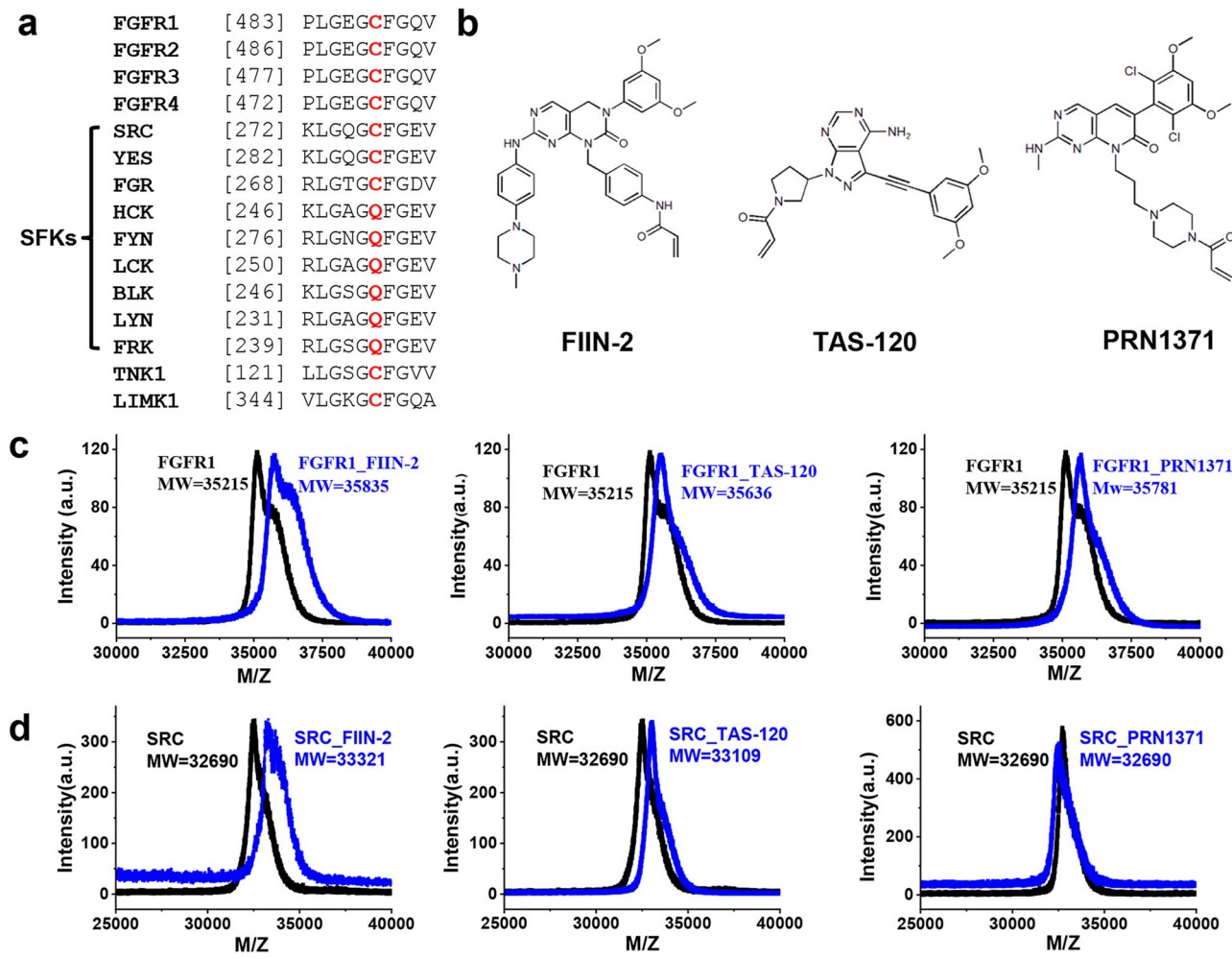

**Fig. 1 Binding modes of FIIN-2, TAS-120 and PRN1371 to FGFR1 and SRC. a** Sequence alignment of the p-loop residues of kinase proteins. The position of the cysteine is indicated in red. **b** Chemical structures of FIIN-2, TAS-120 and PRN1371. **c** MALDI-TOF MS of apo FGFR1 (black) and FGFR1/inhibitor mixture (blue). **d** MALDI-TOF MS of apo SRC (black) and SRC/inhibitor mixture (blue).

Supplementary Fig. 2b)[16]. Some differences between these two complexes are observed: the hydrogen bond between FIIN-2 and Cys277 can only be observed in the SRC/FIIN-2 complex (Fig. 5a); whereas the hydrogen bonds between FIIN-2 and Glu475 can only be observed in FGFR4/FIIN-2 complex (Fig. 5b). In addition, the αC-helix of SRC moves away from the hydrophobic pocket in SRC/FIIN-2 complex, suggesting an inactive conformation, whereas the αC-helix of FGFR4 exhibits active conformation and makes extensive hydrophobic interaction and Van der Waals contacts with FIIN-2 (Fig. 5, Supplementary Fig. 2).

In the SRC/TAS-120 structure, TAS-120 forms hydrogen bonds with Met341 and Thr338 or Glu339 (Fig. 5c and Supplementary Fig. 3a), similar to the binding mode of TAS-120 with FGFR1 (PDB ID: 6MZW) (Fig. 5d and Supplementary Fig. 3b)[22]. The electron density for a region of p-loop (amino acids 274-277, including Cys277) is poor, suggesting that this region is flexible in the structure. The presence of a disordered P-loop was also observed in the structure of TAS-120/FGFR1 complex[22]. Based on our mass spectrometry data, we believe that the covalent bond is present. The αC-helix of SRC also exhibits an inactive conformation in the SRC/TAS-120 structure (Supplementary Fig. 3c). TAS-120 is stabilized by extensive water-mediated interactions with residues Lys514, Glu531 and Asp641 in the FGFR1/TAS-120 structure (2.2 Å) (Supplementary Fig. 3d),

while little water molecules are observed in the SRC/TAS-120 structure owing to a lower resolution (2.7 Å). Overall, these differences may be the reason why TAS-120 show weaker potency to inhibit SRC relative to FGFR kinases.

**Structural basis for FGFR targeting by PRN1371.** Since the structure of FGFR/PRN1371 complex has not been solved, we determined the structure of FGFR4/PRN1371 complex at a resolution of 2.3 Å (Supplementary Table 1). PRN1371 forms two hydrogen bonds with Ala553, and forms a hydrogen bond with Asp630 (Fig. 5e and Supplementary Fig. 4a). Moreover, PRN1371 interacts with Lys503 and Asp630 through water-mediated hydrogen bonds (Supplementary Fig. 4b). A region of p-loop (amino acids 476-479, including Cys477) and the covalent bond have poor density in the structure, suggesting that this region is flexible in the structure. Based on mass spectrometry data, we believe that the covalent is present.

To investigate the mechanism by which PRN1371 is ineffective against SRC, we superposed our structure of FGFR4/PRN1371 complex with a SRC structure (Fig. 5f). Structural analysis shows that SRC has a slightly smaller pocket compared with FGFR4 (Supplementary Fig. 5), which might result from the discrepant pocket residues and the difference in overall backbone between FGFR and SRC (RMSD for Cα being about 1 Å). Moreover, the two chlorine atoms of PRN1371 make the

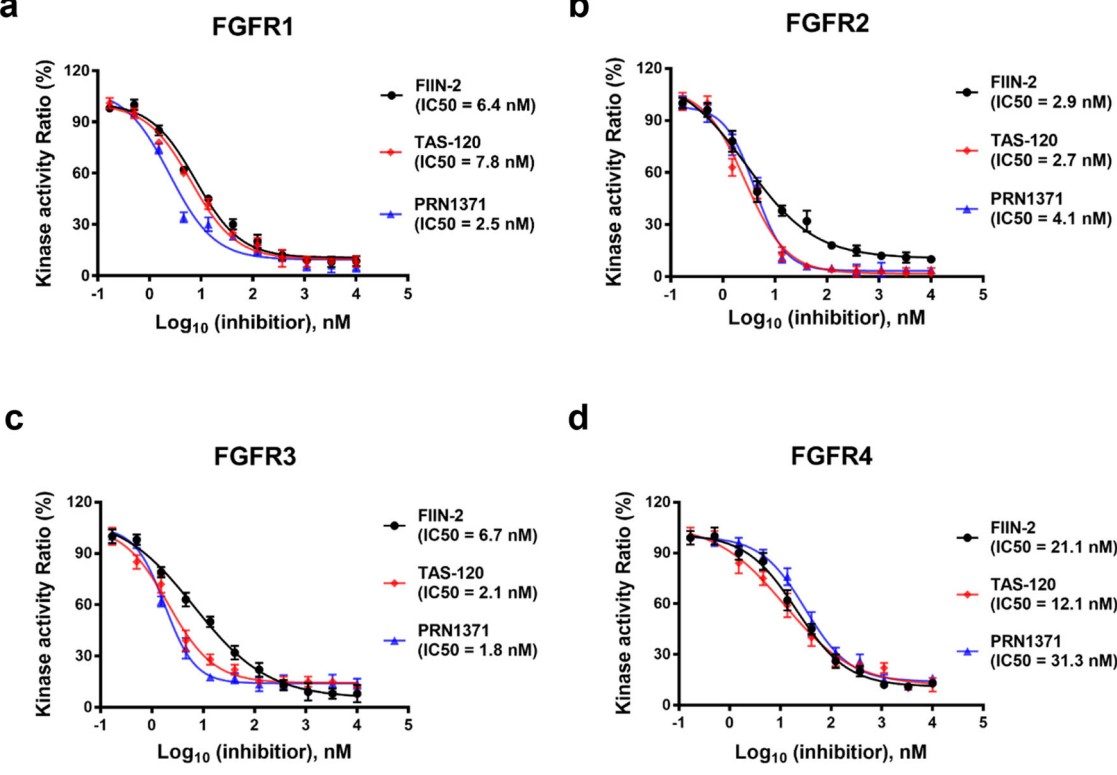

**Fig. 2 Inhibitory potency of FIIN-2, TAS-120 and PRN1371 against FGFRs.** The kinase assays of inhibitors FIIN-2 (black circles), TAS-120 (red diamonds) and PRN1371 (blue triangles) against FGFR1 (**a**), FGFR2 (**b**), FGFR3 (**c**), FGFR4 (**d**) were analyzed using an in vitro kinase assay kit as described in the Materials and Methods. Data are represented as mean of $n = 3$ independent experiments.

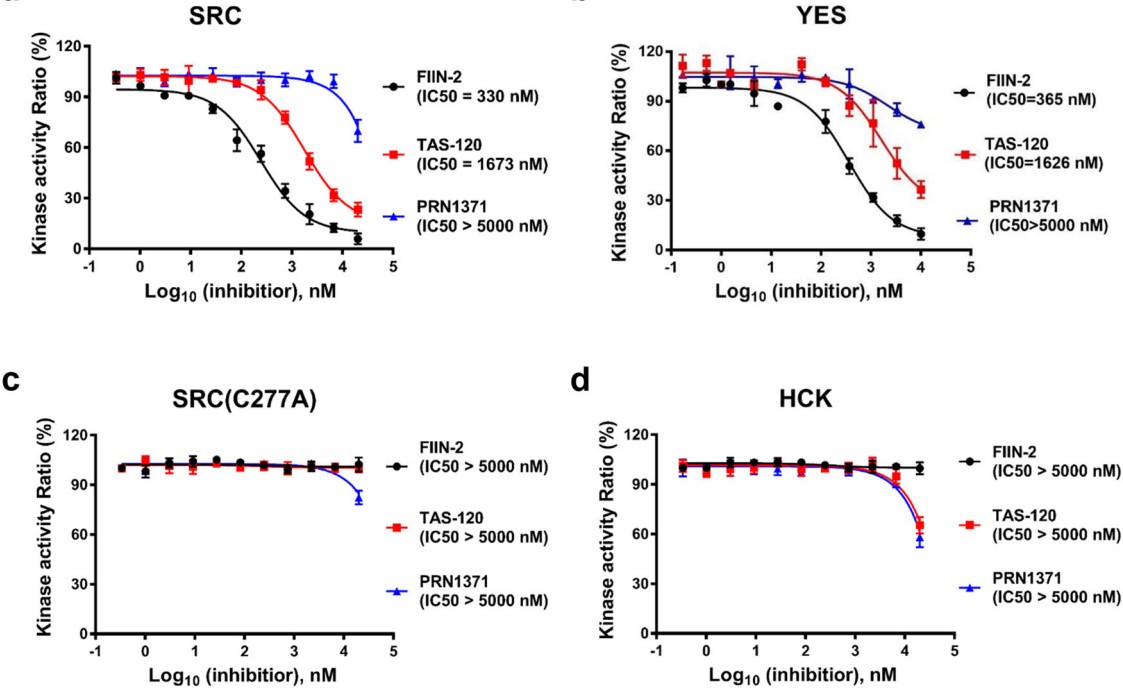

**Fig. 3 Inhibitory potency of FIIN-2, TAS-120 and PRN1371 against SFKs.** The kinase assays of inhibitors FIIN-2 (black circles), TAS-120 (red diamonds) and PRN1371 (blue triangles) against SRC (**a**), YES (**b**), SRC(C277A) (**c**) and HCK (**d**) were performed using an in vitro kinase assay kit as described in the Materials and Methods. Data are represented as mean of $n = 3$ independent experiments.

| IC$_{50}$, nM | | | | | | | | |
|---|---|---|---|---|---|---|---|---|
| kinase activity | FGFR1 | FGFR1 (V561M) | FGFR2 | FGFR2 (V564F) | FGFR3 | FGFR3 (V555M) | FGFR4 | FGFR4 (V550L) |
| FIIN-2 | 6.4 | 216 | 2.9 | 276 | 6.7 | 97 | 21.0 | 255 |
| TAS-120 | 7.8 | 224 | 2.7 | 52 | 2.1 | 110 | 12.1 | 90 |
| PRN1371 | 2.5 | 84 | 4.1 | >1000 | 1.8 | 583 | 31.1 | >1000 |

**Table 1 The IC$_{50}$ of FIIN-2, TAS-120 and PRN1371 to gatekeeper mutants of FGFRs.**

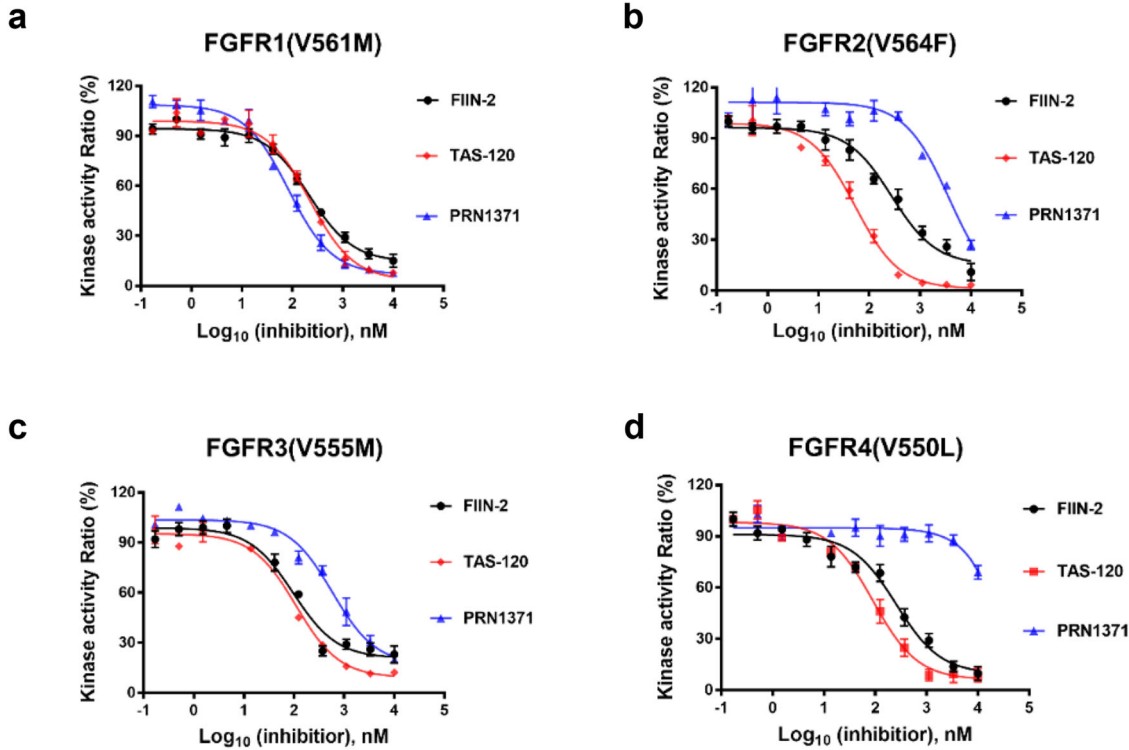

**Fig. 4 Potency of FIIN-2, TAS-120 and PRN1371 against FGFR gatekeeper mutants.** The kinase assays of inhibitors FIIN-2 (black circles), TAS-120 (red diamonds) and PRN1371 (blue triangles) against FGFR1(V561M) (**a**), FGFR2(V564F) (**b**), FGFR3(V555M) (**c**), FGFR(V550L) (**d**) were performed using an in vitro kinase assay kit as described in the Materials and Methods. Data are represented as mean of $n = 3$ independent experiments.

dichloro-dimethoxyphenyl ring less rotationally flexible. Therefore, PRN1371 could clash with Ile335 and Lys295 of SRC (Val548 and Lys503 in FGFR4), preventing PRN1371 from binding to SRC protein.

**Structural basis of drug resistance**. Our kinase assay data reveal that PRN1371 is the least potent inhibitor towards the gatekeeper mutants except in the case of the FGFR1(V561M). To understand the molecular mechanism of drug resistance, we built several structural models and compared the possible binding modes by different inhibitors. Superposition of the structures of FGFR in complex with these inhibitors indicated that the dichloro-dimethoxyphenyl ring of PRN1371 formed very close contacts with the gatekeeper residue Val550 (Fig. 5e, Supplementary Fig. 6), whereas the dimethoxyphenyl ring of FIIN-2 and TAS-120 have more space away from Val550.

When PRN1371 was docked to FGFR1 and the gatekeeper mutant, respectively, the best docking model of PRN1371 with FGFR1(V561M) has a similar predicted binding energy ($-9$ Kcal·mol$^{-1}$) to that of wild-type FGFR1 ($-9.3$ Kcal·mol$^{-1}$); however, the best docking model of PRN1371 with FGFR1(V561F) got a much worse predicted binding energy ($-3.3$ Kcal·mol$^{-1}$) (Fig. 6 and Table 2). It seems that the flexible side chain of methionine can be compatible with the dichloro-dimethoxyphenyl

ring of PRN1371, while substitution with phenylalanine will lead to less interactions between PRN1371 and FGFR1(V561F) (Supplementary Fig. 7). It is unexpected that the methionine of FGFR3(V555M) is not well tolerated by PRN1371 in the kinase assay, which may be some difference between the two kinase. For FIIN-2 and TAS-120, the predicted binding energy of the best docking model for FGFR1 (V561M) or FGFR1 (V561F) was slightly reduced compared to the wild type (Supplementary Figs. 8 and 9). These data are consistent with the kinase assays.

## Discussion

Pan-FGFR covalent inhibitors targeting a unique p-loop cysteine offer promising cancer therapeutic approaches, due to their potential to be highly selective. Among the 9 kinases that share the p-loop cysteine, 4 kinases belong to FGFR family and 3 kinases belong to SRC family[24]. In this study, we compare the selectivity among three covalent inhibitors, FIIN-2, TAS-120, and PRN1371. TAS-120 and PRN1371 are currently in clinical trials[20]. A first-inhuman, phase 1 dose-escalation trial has reported that TAS-120 has shown promising activity in pretreated patients with advanced solid tumors, while the most common adverse events were hyperphosphatemia, diarrhea and constipation[25].

Among these inhibitors, FIIN-2 and TAS-120 form covalent adducts with SRC, and inhibit the kinase activity of SRC and YES

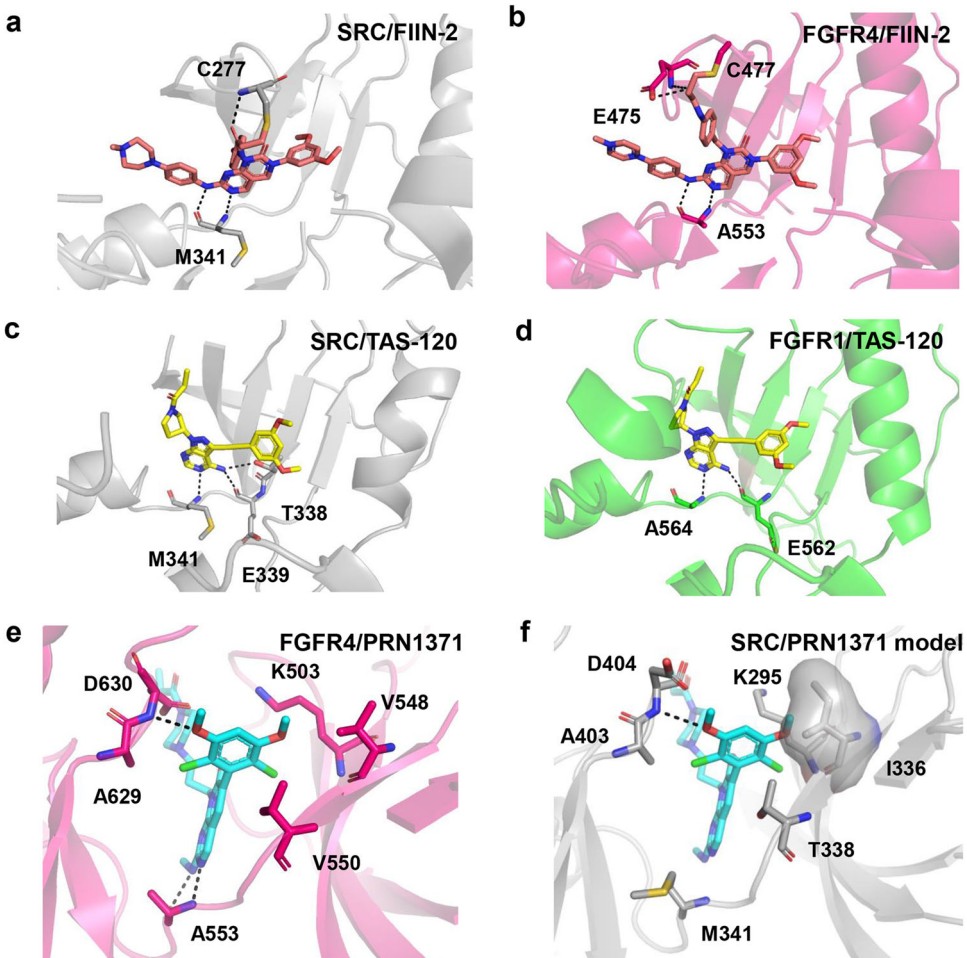

**Fig. 5 Structural basis for SRC and FGFRs targeting by FIIN-2, TAS-120 and PRN1371.** The detailed interactions are shown in the structures of SRC/FIIN-2 (**a**), FGFR4/FIIN-2 (PDB ID: 4QQC) (**b**), SRC/TAS120 (**c**), FGFR1/TAS-120 (PDB ID: 6MZW) (**d**), FGFR4/PRN1371 (**e**) and modeled SRC/PRN1371 (**f**). The modeled SRC/PRN1371 structure was acquired by a superposition of SRC/FIIN-2 structure with the FGFR4/PRN1371 structure. Hydrogen bond is shown as black dotted lines. SRC is colored gray; FGFR1 is colored green; FGFR4 is colored pink; FIIN-2 is shown as salmon stick; TAS-120 is shown as yellow stick; PRN1371 is shown as cyan stick.

proteins. Although their potency against SRC and YES are much weaker than that of FGFR proteins, FIIN-2 and TAS-120 may have off-target issues due to the formation of the covalent adducts. Notably, PRN1371 seems to be a more selective covalent inhibitor. Our mass spectrometry data show that PRN1371 does not form covalent adduct with SRC, and our kinase assays indicate that it has little effect on the kinase activity of SRC and YES proteins. Previous studies show that Bcr-Abl inhibitor Dasatinib could also inhibit SRC family kinases, and their inhibitions may cause adverse events such as skin eruption, fluid retention, skin or hair depigmentation, and pulmonary arterial hypertension (PAH)[21,26]. Since FIIN-2 and TAS-120 can also target SRC and YES proteins, adverse events associated with inhibition of SFK proteins should be monitored.

One of obstacles in the development of kinase inhibitor is the drug resistance. Mutation of the "gatekeeper" residue located in the hinge region of the kinases ATP-binding pocket is a common mechanism of acquired resistance. FGFR2 gatekeeper mutation V564F has been reported in three intrahepatic cholangiocarcinoma (ICC) patients treated with Pemigatinib[27]. FGFR4 gatekeeper mutations V550L/V550M have been reported in patients with hepatocellular carcinoma after treatment with BLU-554[28]. In our study, we compared the ability to overcome gatekeeper mutations among the three pan-FGFR covalent inhibitors. All

three inhibitors show decreased potency against gatekeeper mutants than that against wild-type FGFR proteins. Of the three inhibitors, TAS-120 exhibits the highest overall potency against FGFR gatekeeper mutants, while PRN1371 is the least potent.

It seems that the two chloride atoms of PRN1371 play key roles in the resistance to gatekeeper mutations. The two chloride atoms make the dichloro-dimethoxyphenyl ring bulkier and less rotationally flexible than the dimethoxyphenyl ring, and one of the chloride atoms forms favorable contact with the small gatekeeper residue valine. The phenomenon is also observed in the structures of FGFRs in complex with BGJ398 and FIIN-1[29,30]. These inhibitors, PRN1371, BGJ398 and FIIN-1, all show little potency against FGFR gatekeeper mutations[16]. This may be partly attributed to steric clashes between the dichloro-dimethoxyphenyl ring and a larger side chain of gatekeeper mutant such as phenylalanine. Compared to PRN1371, the dimethoxybenzene ring of FIIN-2 and TAS-120 has a larger space near the gatekeeper. TAS-120 even has a less bulky linker to its core structure, giving its dimethoxyphenyl ring great rotatory flexibility to accommodate the hydrophobic pocket of FGFRs.

Notably, the potency against gatekeeper mutations not only depends on the small-molecule inhibitor, but also depends on which protein is the target. Although the four members of FGFR family share high structural similarity, these three

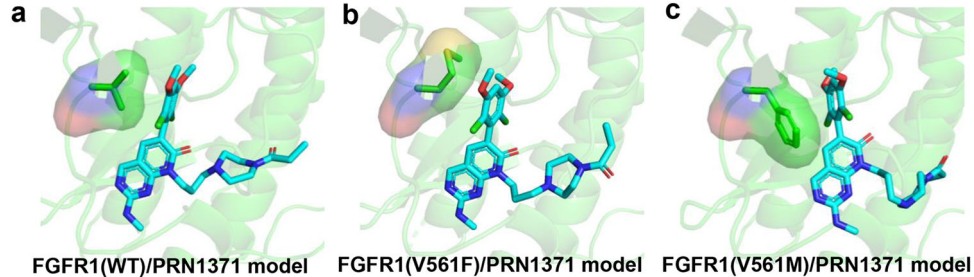

**Fig. 6 Structural models of FGFR1 gatekeeper mutants with PRN1371.** Docking model of PRN1371 to FGFR1(WT) (**a**), FGFR1(V561M) (**b**) and FGFR1(V561F) (**c**). Molecular docking was performed by AutoDock Tools program.

**Table 2 Predicted binding energies of FGFR1 gatekeeper mutants with PRN1371.**

|  | WT | V561M | V561F |
|---|---|---|---|
| Predicted Binding Energy (Kcal·mol$^{-1}$) | −9.3 | −9.0 | −3.3 |

inhibitors exhibit different abilities against different FGFR gatekeeper mutants. For example, PRN1371 is most effective against FGFR1(V561M) mutant, FIIN-2 is most potent against FGFR3(V555M) mutant, and TAS-120 is most effective against FGFR2(V564F) and FGFR4(V550L) mutants. Interestingly, PRN1371 is effective against FGFR1(V561M) mutant, but is ineffective against FGFR4(V550L) mutant. Our structural models predict that the mutated FGFR4 Leu550 will form steric clash with PRN1371, and prevent PRN1371 from binding to FGFR4(V550L) mutant. Moreover, PRN1371 is effective against FGFR1(V561M) mutant, but is ineffective against FGFR3(V555M) mutant, which maybe resulting from the difference between the two kinase.

In order to achieve better clinical benefits, the development of next-generation FGFR covalent inhibitors with high selectivity and the ability to overcome gatekeeper mutations is needed. In this study, we show that PRN1371 is the most selective among the three pan-FGFR covalent inhibitors. Although all three inhibitors retain some potencies against FGFR gatekeeper mutants, their potencies against FGFR gatekeeper mutants are much weaker than that against wild-type proteins. We anticipate that our studies will inspire the development of next-generation FGFR covalent inhibitors with high potency, high selectivity, and stronger ability to overcome gatekeeper mutations.

## Methods

**Protein expression and purification.** FGFRs, SRC and HCK were prepared as previously described[31–33]. Briefly, the kinase domain of human FGFR1 (residues 456−765), FGFR2 (residues 453−770), FGFR3 (residues 450−758), FGFR4 (residues 445−753), chicken SRC (residues 251−533) and human HCK (residues 247−526) were respectively cloned into a modified pET28a expression vector with a N-terminal 6×His tag followed by a PreScission or TEV cleavage site. The mutants, FGFR1(V561M), FGFR2(V564F), FGFR3(V555M), FGFR4(V550L), FGFR4(C477A), SRC(C277A), were introduced by PCR using primers with the desired mutations. The primers were synthesized by Tsingke Biotechnolog Co., Ltd., and the sequences were listed in Supplementary Table 2. Then plasmid was transfected into *E. coli* BL21 (DE3) cells and co-expressed with YOPH to get the non-phosphorylated proteins[31]. Cells were grown in LB at 37 °C and induced for 16 h with 0.5 mM IPTG at 18 °C. Cells were harvested by centrifugation for next purification. All steps of purification were performed at 4 °C. Proteins were first purified by Ni-NTA affinity chromatography, and then 6×His tag was removed by PreScission or TEV protease. SRC(C277A) was purified as His-tagged fusion proteins. Anion exchange chromatography (Mono Q) and size exclusion chromatography (SEC) were exploited for further purification. The purified FGFRs proteins were stored in the buffer containing 20 mM tris-HCl pH 8.0, 150 mM NaCl, and 1 mM TCEP, while SRC and HCK proteins in the buffer containing 20 mM tris-HCl pH 8.0, 150 mM NaCl, 5% (v/v) glycerol, and 1 mM TCEP. The

purified proteins were concentrated to about 5–15 mg/mL and stored at −80 °C for subsequent studies. Human YES protein was purchased from Carna company (Part No #08-175).

**MALDI-TOF-MS.** Matrix-assisted laser desorption/ionization time-of-flight mass spectrometry (MALDI-TOF-MS) was preformed to determine the molecular weight of samples[32]. Sinapic acid (Sigma Part No #85429) was used as the matrix on the MALDI-TOF apparatus (AB SCIEX 5800). Linear positive ion mode, containing the suitable detection range of molecular weight, was selected as the detection method. Proteins were incubated with three inhibitors at 1:2 molar ratio overnight at 4 °C and then the salt was removed by ultrafiltration. Desalting samples (<1 mg/mL) were first mixed with matrix solution (20 mg/ml) at a volume ratio of 1:1, then pipetted onto the target plate and dried at room temperature. Mass spectrogram was generated using FlexAnalysis and Origin.

**Kinase inhibition assay.** The ADP-Glo$^{TM}$ Kinase Assay Kit (Part No #V9101) was purchased from Promega. The experiments were performed according to the manufacturer's instructions. All kinase inhibition assays were performed with optimized kinase assay buffer consisting of 40 mM Tris-HCl (pH ranging from 7.5 to 8.0), 20 mM MgCl$_2$, 20 mM NaCl, 0.1 mg/mL BSA, 1 mM TCEP, and 4% DMSO[34]. First, inhibitors FIIN-2, TAS-120 and PRN1371 in triple dilutions (ranging from 0.01 nM to 25 μM) were incubated with kinases (0.1 μM) in white opaque 384-well plates for half an hour at room temperature, respectively. Second, 5×ATP plus poly (4:1 Glu, Tyr) peptides (abcam, Part No #ab204877) were added to start the kinase reactions. The final reaction consisted of 0.04 μM kinase, 50 μM Tyr4 peptide and 10 μM ATP in kinase buffer. Then the reactions were terminated by the addition of stop buffer ADP-Glo after half an hour of incubation. The produced ADP was completely converted to ATP after 40 min of incubation. Finally, fluorescence was measured on a multimode plate reader (Perkin-Elmer) after the addition of detection reagent for half an hour to one hour. IC$_{50}$ values were determined using a three-parameter log [Inhibitor] versus response model in GraphPad Prism software.

**Crystallization.** Crystals were obtained by the hanging drop vapor diffusion method. Protein-inhibitor complexes were prepared by mixing protein (5–15 mg/ml) with inhibitor at 1:2 molar ratio and incubated on ice for 30 min. Crystals of SRC/FIIN-2 and SRC/TAS-120 complexes were grown in 3 days at 18°C using a well solution containing 0.1 M MES (pH 6.4), 2% glycerol, 8% PEG 4000, 50 mM sodium acetate, 10 mM MgCl$_2$. Crystals of FGFR4/PRN1371 complex were grown under 0.1 M Bis-Tris (pH 4.5), 0.2 M Li$_2$SO$_4$, 18% PEG 3350 at 4 °C. Prior to diffraction experiments, crystals were cryoprotected by supplementing the mother liquor with 20% glycerol, then cooled in liquid nitrogen.

**X-ray data collection, data processing and structures solution.** SRC/FIIN-2 and SRC/TAS-120 crystal datasets were collected at the BL17U[35] and BL19U1[36] beamlines of Shanghai Synchrotron Radiation Facility (SSRF), FGFR4/PRN1371 crystal data were collected using an in-house MicroMax-007 x-ray generator equipped with VariMax HR optics (Rigaku, Japan). The diffraction data were processed using HKL2000[37]. The structures were solved by molecular replacement with phaser from PHENIX package[38] using the previously solved SRC/Ruxolitinib structure (PDB ID: 4U5J)[31] as the search model for SRC/FIIN-2 and SRC/TAS-120, and FGFR4/ponatinib structure (PDB ID: 4QRC)[39] as the search model for FGFR4/PRN1371. The ligands were drawn by ChemDraw software and the models were building on the website http://davapc1.bioch.dundee.ac.uk/cgi-bin/prodrg/submit.html. Phenix.ligandfit was run to place the ligand and the correlation coefficient (CC) value was above 0.7. The best-fitting ligand was added to the structure. Then structures were refined with phenix.refine and model building was performed using WinCoot[40]. Structural graphics were drawn by PyMOL[41]. Data collection and structure refinement statistics are presented in Supplementary Table 1.

**Molecular docking**. Computational docking was performed to predict the binding of FIIN-2, TAS-120 and PRN1371 to FGFRs gatekeeper mutations. The structures of FGFR gatekeeper mutations were generated by substitution of gatekeeper residues on the basis of FGFR1/TAS-120 (PDB:6MZW). Polar hydrogens, Gasteiger charges and rotatable bonds to the inhibitors were assigned by AutoDock Tools program[42]. The docking simulations were carried out with rigid FGFR and a flexible ligand. A docking grid with the dimensions of 40*40*40 (TAS-120) or 52*50*50 (FIIN-2 and PRN1371) points in the x-, y-, and z-axis directions was built, which encompassed the entire ligand-binding clefts.

**Reporting summary**. Further information on research design is available in the Nature Research Reporting Summary linked to this article.

## Data availability

The coordinates and structure factors are deposited in the Protein Data Bank under the accession codes 7D57 (SRC/FIIN-2 complex), 7D5O (SRC/TAS-120 complex), 7F3M (FGFR4/PRN1371 complex). Validation reports are available as Supplementary Data 1. All other relevant data supporting the key findings of this study are available within the article and its Supplementary Information files. A reporting summary is available as a Supplementary Information file.

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

## Acknowledgements

This work was supported by the National Natural Science Foundation of China (grants 81974074 and 82172654 to Y.C.; grant 31900880 to H.W.), China Postdoctoral Science Foundation (2019M652805), Science and Technology Planning Project of Hunan Province (2018TP1017 and 2021RC4012), Natural Science Foundation of Hunan Province (2021JJ40961). We thank the staffs from BL17U/BL19U1 beamline of National Facility for Protein Science in Shanghai (NFPS) at Shanghai Synchrotron Radiation Facility, for assistance during data collection.

## Author contributions

L.Q. and X.C. performed experiments; M.G., S.D., L.J., J.L. and S.Y. performed data collection and structure determination. L.Q., X.C., Z.Z., H.W., Z.C. and Y.C. analyzed the data. L.Q., X.C., H.W. and Y.C. prepared the manuscript.

## Competing interests

The authors declare no competing interests.
