## [Peer Review File · Communications Chemistry]

This manuscript has been previously reviewed at another Nature Portfolio journal. This document only contains reviewer comments and rebuttal letters for versions considered at Communications Chemistry.

Reviewers' comments:

Reviewer #1 (Remarks to the Author):

This new version of a previous submission is greatly improved and addresses most of the issues highlighted in the earlier review. The structures and analysis presented are new information although some discussion sections need reworking and the conclusions around gatekeeper mutations here are not particularly novel (many would make similar conclusions from inspection of previous structures - in relation to Cl substituted rings). Collectively the data and analysis (in particular the solid kinase assay and mass spectrometry work) is worth publishing and adds to the literature. Scientists working in the field of kinase drug discovery will find the new structures and data useful.

I have provided annotation in the submission PDF document - please see specific comments to address therein.

Additional notes:

1. Supplemental Figure 2 (c). Superimposition of protein but not ligand. Ligand inclusion would be helpful to show similarities in binding modes. similarly for Supp Fig 3.
2. Supp Fig 8. FIIN-2 model has a planar ring in the model where it should not. This could impact on the theoretical model produced (even though that part is likely extended into solvent space)
3. Concerns in the models submitted to the PDB. It is difficult to assess models from the PDB validation report alone - always better to include structures themselves in journal submission for proper review. This is critically important when the main thrust of the article is to show ligand interactions with a target protein.

In the TAS-120 structure - for DRG B 601 a methyl group of the dimethoxy phenyl is not modelled appropriately. This should be fixed. The methyl group should be directed towards the green section indicated in the maps.

In the PRN1371 structure, Cys SG to GX3 C26 close contact at 2.29 angstrom - this is the covalent bond, right? In which case it is too long I would have thought? How long should a C-S bond be? Is a link command being used to constrain the distance? Another distance ARG NH2 to GX3 C20 2.51 - this looks like too close to me. Sum of a C and N vdW radius is closer to 3.2 angstrom. This suggests that the ligand could be modelled more appropriately.

In the FIIN-2 structure again several close contacts - I would certainly delete the waters involved where H-bonds appear too short. There are also protein-protein contacts that appear too short.

I think the structures need to be reconsidered and remodeled to remove the too close contacts with stronger restraints used and or deletion of poor features like water molecules.

Structural insights into the potency and selectivity of covalent pan-FGFR inhibitors

Lingzhi Qu^{1,4}, Xiaojuan Chen^{1,4}, Hudie Wei^{1,*}, Ming Guo¹, Shuyan Dai¹, Longying Jiang^{1,2},
Jun Li¹, Sitong Yue¹, Zhuchu Chen¹, Yongheng Chen^{1,3,*},†

¹Department of Oncology, NHC Key Laboratory of Cancer Proteomics, State Local Joint Engineering Laboratory for Anticancer Drugs, Xiangya Hospital, Central South University, Changsha, Hunan 410008, China.

²Department of Pathology, Xiangya Hospital, Central South University, Changsha, Hunan 410008, China

³National Clinical Research Center for Geriatric Disorders, Xiangya Hospital, Central South University, Changsha, Hunan 410008, China.

⁴These authors contribute equally to this work.

†Lead contact

*Corresponding author. Email: yonghenc@163.com; hudiewei18@163.com

18 **Abstract**

[revised manuscript text omitted]
E4869-4877, doi:10.1073/pnas.1403438111 (2014).
Duan, Y., Chen, L., Chen, Y. & Fan, X. G. c-Src binds to the cancer drug Ruxolitinib with an active
conformation. *PLoS one* **9**, e106225, doi:10.1371/journal.pone.0106225 (2014).
Zhou, Z. *et al.* Characterization of FGF401 as a reversible covalent inhibitor of fibroblast growth factor
receptor 4. *Chem. Commun.* **55**, 5890-5893, doi:10.1039/c9cc02052g (2019).
Guo, M. *et al.* Characterization of ibrutinib as a non-covalent inhibitor of SRC-family kinases. *Bioorg.*
*Med. Chem. Lett.* **34**, 127757, doi:10.1016/j.bmcl.2020.127757 (2021).
Wu, D. *et al.* LY2874455 potently inhibits FGFR gatekeeper mutants and overcomes mutation-based
resistance. *Chem Commun (Camb)* **54**, 12089-12092, doi:10.1039/c8cc07546h (2018).
Wang, Q.-S. *et al.* Upgrade of macromolecular crystallography beamline BL17U1 at SSRF. *Nuclear*
*Science and Techniques* **29**, 68, doi:10.1007/s41365-018-0398-9 (2018).
Zhang, W.-Z. *et al.* The protein complex crystallography beamline (BL19U1) at the Shanghai
Synchrotron Radiation Facility. *Nuclear Science and Techniques* **30**, 170, doi:10.1007/s41365-019-
0683-2 (2019).
Otwinowski, Z. & Minor, W. Processing of X-ray diffraction data collected in oscillation mode. *Methods*
*Enzymol.* **276**, 307-326 (1997).
Liebschner, D. *et al.* Macromolecular structure determination using X-rays, neutrons and electrons:
recent developments in Phenix. *Acta crystallographica. Section D, Structural biology* **75**, 861-877,
doi:10.1107/S2059798319011471 (2019).
Huang, Z. *et al.* DFG-out mode of inhibition by an irreversible type-1 inhibitor capable of overcoming
gate-keeper mutations in FGF receptors. *ACS chemical biology* **10**, 299-309, doi:10.1021/cb500674s
(2015).

Emsley, P., Lohkamp, B., Scott, W. G. & Cowtan, K. Features and development of Coot. *Acta*
 *crystallographica. Section D, Biological crystallography* **66**, 486-501, doi:10.1107/S0907444910007493
 (2010).
 Schrodinger, LLC. *The PyMOL Molecular Graphics System, Version 1.8* (2015).
 Morris, G. M. *et al.* AutoDock4 and AutoDockTools4: Automated docking with selective receptor
 flexibility. *J Comput Chem* **30**, 2785-2791, doi:10.1002/jcc.21256 (2009).

 **Fig. 1. Binding modes of FIIN-2, TAS-120 and PRN1371 to FGFR1 and SRC.** (a) Sequence
 alignment of the p-loop residues of kinase proteins. The position of the cysteine is indicated in
 red. (b) Chemical structures of FIIN-2, TAS-120 and PRN1371. (c) MALDI-TOF MS of apo
 FGFR1 (black) and FGFR1/inhibitor mixture (blue). (d) MALDI-TOF MS of apo SRC (black)
 and SRC/inhibitor mixture (blue).

**Fig. 2. Inhibitory potency of FIIN-2, TAS-120 and PRN1371 against FGFRs.** The kinase
 assays of inhibitors FIIN-2, TAS-120 and PRN1371 against FGFR1(a), FGFR2(b), FGFR3(c),
 FGFR4(d). Inhibitors activity against FGFRs were analyzed using an *in vitro* kinase assay kit
 according to the manufacturer's instructions. Data are represented as mean \pm SD of values from
 3 independent experiments.

**Fig. 3. Inhibitory potency of FIIN-2, TAS-120 and PRN1371 against SFKs.** The kinase

assays of inhibitors FIIN-2, TAS-120 and PRN1371 against SRC (a), YES (b), SRC(C277A)

(c) and HCK (d). The *in vitro* SFKs kinase assay was performed as described in the Materials

and Methods. Data are represented as mean \pm SD of values from 3 independent experiments.

a

kinase activity	IC ₅₀ , nM							
	FGFR1	FGFR1 (V561M)	FGFR2	FGFR2 (V564F)	FGFR3	FGFR3 (V555M)	FGFR4	FGFR4 (V550L)
FIIN-2	6.4	216	2.9	276	6.7	97	21.0	255
TAS-120	7.8	224	2.7	52	2.1	110	12.1	90
PRN1371	2.5	84	4.1	>1000	1.8	583	31.1	>1000

**Fig. 4. Potency of FIIN-2, TAS-120 and PRN1371 against FGFR gatekeeper mutants. (a)**

The IC₅₀ of FIIN-2, TAS-120 and PRN1371 to gatekeeper mutants of FGFRs. (b-e) The kinase

assays of inhibitors FIIN-2, TAS-120 and PRN1371 against FGFR1(V561M) (b),

FGFR2(V564F) (c), FGFR3(V555M) (d), FGFR(V550L) (e). Data are represented as mean ±

SD of values from 3 independent experiments.

**Fig. 5. Structural basis for SRC and FGFRs targeting by FIIN-2, TAS-120 and PRN1371.**

The detailed interactions are shown in the structures of SRC/FIIN-2 (a), FGFR4/FIIN-2 (PDB

ID: 4QQC) (b), SRC/TAS120 (c), FGFR1/TAS-120 (PDB ID: 6MZW) (d), FGFR4/PRN1371

(e) and modelled SRC/PRN1371 (f). The modelled SRC/PRN1371 structure was acquired by a

superposition of SRC/FIIN-2 structure with the FGFR4/PRN1371 structure. Hydrogen bond is

shown as black dotted lines. SRC is colored grey; FGFR1 is colored green; FGFR4 is colored

pink; FIIN-2 is shown as salmon stick; TAS-120 is shown as yellow stick; PRN1371 is shown

as cyan stick.

**Fig. 6. Structural models of FGFR1 gatekeeper mutants with PRN1371.** Docking model of
 PRN1371 to FGFR1(WT) (a), FGFR1(V561M) (b) and FGFR1(V561F) (c). (d) Predicted
 binding energies of FGFR1 gatekeeper mutants with PRN1371. Molecular docking was
 performed by AutoDock Tools program.

Reviewers' comments:

Reviewer #1 (Remarks to the Author):

This new version of a previous submission is greatly improved and addresses most of the issues highlighted in the earlier review. The structures and analysis presented are new information although some discussion sections need reworking and the conclusions around gatekeeper mutations here are not particularly novel (many would make similar conclusions from inspection of previous structures - in relation to Cl substituted rings). Collectively the data and analysis (in particular the solid kinase assay and mass spectrometry work) is worth publishing and adds to the literature. Scientists working in the field of kinase drug discovery will find the new structures and data useful.

I have provided annotation in the submission PDF document - please see specific comments to address therein.

The responses for the specific details and comments are listed below

1. On page 6, line 141, I don't think you can conclude this. The resolution of the two structures is very different. You will see many more water molecules in a sub-2 angstrom structure compared to a 2.7 angstrom structure where you will likely see none - even though they may well exist as ordered water molecules. simply a function of resolution. Please add a qualifying comment that the resolutions are different and this impacts on the assessment/comparison of water structure.

Response: Thank you for the insightful comment. We have added this information in the revised manuscript.

2. On page 7, line 162, This title should be changed. The drug resistance is not overcome by any of the drugs considered. the title line should change to "Structural basis of drug resistance". The section does not address any means to overcome drug resistance.

Response: Thanks for the suggestion. We have changed the title.

3. On page 7, line 164, except for the FGFR1 case - so you need to note this - it is the least potent except in the case of the FGFR1 V561M - but then still the worst against the equivalent FGFR3 mutant - this should be discussed in the text.

Response: Thanks for the comment. In the revised manuscript, we have noted this information, and discussed the results about equivalent FGFR3 mutant in the revised manuscript.

4. On page 7, line 164, remove "overcome"

Response: We have removed "overcome" in the revised manuscript.

5. On page 7, line 169, what is the contrast? Are clashes found in the FIIN-2 and TAS-120 structures?

Response: The dimethoxyphenyl ring of FIIN-2 and TAS-120 is relatively distant from Val550 than the dichloro-dimethoxyphenyl ring of PRN1371. There is no clash in the FIIN-2 and TAS-120 structures. We have rephrased the sentence in the revised manuscript.

6. On page 8, line 176, This discussion does not explain the FGFR3 result? where the methionine is not well tolerated.

Response: Thanks for the suggestion. In the revised manuscript, we have added the relative sentences about FGFR3 result in the revised manuscript.

7. On page 9, line 214, reword. The ring is not more rigid. It is both bulkier (with the addition of two relatively large atoms) and likely to be less rotationally flexible. I'm not sure I would consider either of these things are producing a more rigid ring.

Response: Thanks for the suggestion. We have reworded the sentences.

8. On page 9, line 221, No comments on FIIN-2? Has the same ring type as TAS-120? But less space nearby to the ring (TAS-120 has a less bulky linker to its core structure containing the C-C triple bond).

Response: Thanks for the suggestion. In the revised manuscript, we have added the comments on FIIN-2.

9. On page 13, line 297, Home source details?

Response: We have provided the source details in the revised manuscript.

10. Figures 5f, no H-bonds shown here for SRC/PRN model? If there are none then how reliable is that model?

Response: Thanks for the comment. We have added the H-bonds in revised Figure 5f.

Additional notes:

1. Supplemental Figure 2 (c). Superimposition of protein but not ligand. Ligand inclusion would be helpful to show similarities in binding modes. Similarly for Supp Fig 3.

Response: Following the reviewer's suggestion, we have redrawn the Supplemental

Figure 2 (c) and Supplemental Figure 3 (c).

2. Supp Fig 8. FIIN-2 model has a planar ring in the model where it should not. This could impact on the theoretical model produced (even though that part is likely extended into solvent space)

Response: Thanks for the suggestion. In the revised manuscript, the FIIN-2 from SRC/FIIN-2 structure has used to re-build the docking model. We have revised the Supplemental Figure 8 accordingly.

3. Concerns in the models submitted to the PDB. It is difficult to assess models from the PDB validation report alone - always better to include structures themselves in journal submission for proper review. This is critically important when the main thrust of the article is to show ligand interactions with a target protein.

Response: Thanks for the suggestion. We have provided the PDB structures.

4. In the TAS-120 structure - for DRG B 601 a methyl group of the dimethoxy phenyl is not modelled appropriately. This should be fixed. The methyl group should be directed towards the green section indicated in the maps.

In the PRN1371 structure, Cys SG to GX3 C26 close contact at 2.29 angstrom - this is the covalent bond, right? In which case it is too long I would have thought? How long should a C-S bond be? Is a link command being used to constrain the distance? Another distance ARG NH2 to GX3 C20 2.51 - this looks like too close to me. Sum of a C and N vdW radius is closer to 3.2 angstrom. This suggests that the ligand could be modelled more appropriately.

In the FIIN-2 structure again several close contacts - I would certainly delete the waters involved where H-bonds appear too short. There are also protein-protein contacts that appear too short.

I think the structures need to be reconsidered and remodeled to remove the too close contacts with stronger restraints used and or deletion of poor features like water molecules.

Response: Thanks for the suggestion. We have re-refined the structures according to the reviewer's suggestion. All the structural figures and supplementary Table 1 have been revised accordingly.

In the TAS-120 structure, there is clash when the methyl group direct towards the green section (figure shown as follows).

In the FGFR4/PRN1371 structure, we have refined the structure accordingly. Because of poor density of the region of p-loop (amino acids 476-479, including Cys477), the covalent bond (should be about 1.7Å) was not linked in the structure.

In the FIIN-2 structure, we have refined close contact accordingly.

REVIEWERS' COMMENTS:

Reviewer #2 (Remarks to the Author):

The authors have adequately addressed the concerns of the reviewer and I support publication of the revised manuscript in Communications Chemistry.